# TDP2 suppresses chromosomal translocations induced by DNA topoisomerase II during gene transcription

Fernando Gómez-Herreros[1,2,3], Guido Zagnoli-Vieira [1], Ioanna Ntai [1], María Isabel Martínez-Macías[1], Rhona M. Anderson [4], Andrés Herrero-Ruíz[1,5] & Keith W. Caldecott[1]

DNA double-strand breaks (DSBs) induced by abortive topoisomerase II (TOP2) activity are a potential source of genome instability and chromosome translocation. TOP2-induced DNA double-strand breaks are rejoined in part by tyrosyl-DNA phosphodiesterase 2 (TDP2)-dependent non-homologous end-joining (NHEJ), but whether this process suppresses or promotes TOP2-induced translocations is unclear. Here, we show that TDP2 rejoins DSBs induced during transcription-dependent TOP2 activity in breast cancer cells and at the translocation 'hotspot', *MLL*. Moreover, we find that TDP2 suppresses chromosome rearrangements induced by TOP2 and reduces TOP2-induced chromosome translocations that arise during gene transcription. Interestingly, however, we implicate TDP2-dependent NHEJ in the formation of a rare subclass of translocations associated previously with therapy-related leukemia and characterized by junction sequences with 4-bp of perfect homology. Collectively, these data highlight the threat posed by TOP2-induced DSBs during transcription and demonstrate the importance of TDP2-dependent non-homologous end-joining in protecting both gene transcription and genome stability.

---

[1] Genome Damage and Stability Centre, School of Life Sciences, University of Sussex, Falmer, Brighton BN1 9RQ, UK. [2] Instituto de Biomedicina de Sevilla (IBiS), Hospital Virgen del Rocío-CSIC-Universidad de Sevilla, Seville 41013, Spain. [3] Departamento de Genética, Universidad de Sevilla, Seville 41080, Spain. [4] Institute of Environment, Health and Societies, Brunel University London, Uxbridge UB8 3PH, UK. [5] Present address: Centro Andaluz de Biología Molecular y Medicina Regenerativa (CABIMER), CSIC-Universidad de Sevilla, Seville 41092, Spain. Correspondence and requests for materials should be addressed to F.G-H. (email: fgomezhs@us.es) or to K.W.C. (email: k.w.caldecott@sussex.ac.uk)

Tumorigenesis is a multistage process that involves the accumulation of genetic changes. Cytogenetic aberrations are common features of neoplasia and have a key impact in cancer development. Whereas gross chromosomal abnormalities such as duplications, deletions, and chromosome loss or gain are commonly involved in late stages of tumor development, reciprocal translocations represent single events that contribute to initial stages of cellular disarray. The molecular mechanisms that underlie the formation of reciprocal translocations are of great interest, because they can result chimeric proteins and/or deregulated transcription programmes that can drive oncogenesis in a variety of solid tumors and leukemias[1–3]. A number of reciprocal translocations involve partner genes that are co-located in a common transcription factory such as *IgH* and *myc* in Burkitt´s lymphoma[4], *TMPRSS2* and *ERG* in prostate cancer[5], and mixed lineage leukemia (*MLL*), *AF4* and *AF9* in leukemia[6, 7]. Consistent with the idea that spatial proximity is an important factor in reciprocal translocation formation[8], translocations arise from DNA double-strand breaks (DSBs) arising in nearby genes[2]. Nonetheless, the molecular mechanisms that underlie DSB induction and gene fusion during the translocation of actively transcribed genes are unclear and controversial. Indeed, some studies suggest that DSBs that drive translocations are obligate and/or are 'programmed' intermediates of some transcription events[9–12].

One class of enzymes implicated in DSB induction during gene expression are DNA topoisomerases. Topoisomerases, are enzymes that remove torsional stress in DNA by introducing transient DNA breaks and are involved in a wide variety of nuclear processes including transcription and DNA replication[13]. Type II topoisomerases (e.g. TOP2) pass one DNA duplex through another via a DSB intermediate known as the 'cleavage complex'; within which the topoisomerase has cleaved both strands of DNA and is covalently linked to the 5'-terminus of the DSB via a phosphotyrosyl bond[14]. The cleavage complex is normally transient, because the topoisomerase reseals the break at the end of its catalytic cycle. However, cleavage complexes are the targets of a class of anti-tumor agents such as etoposide that act as topoisomerase 'poisons', resulting in the formation of 'abortive' cleavage complexes that require removal by DSB repair.

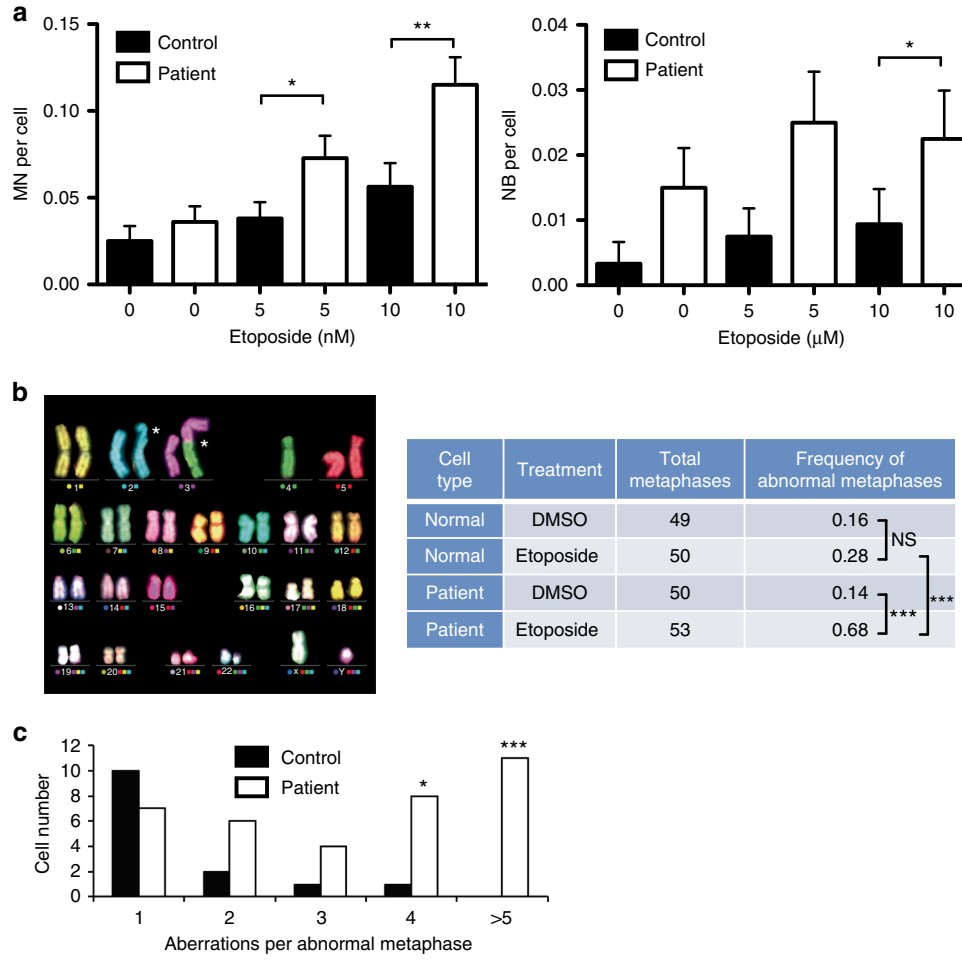

**Fig. 1** TDP2 suppresses TOP2-induced genome instability in human cells. **a** Micronuclei (MN, *left*) and nucleoplasmic bridges (NB, *right*) were quantified in control and TDP2 patient lymphoblastoid cells following continuous treatment (24 h) with the indicated concentration of etoposide. Data are the average (±s.e.m.) of a minimum of 300 cells from three independent experiments. Statistical significance was determined by *T*-test (**P* < 0.05, ***P* < 0.01). **b** M-FISH karyotyping of control and TDP2 patient lymphoblastoid cells following continuous treatment with 50 nM etoposide for 20 h. *Left*, example of a patient metaphase depicting a chromatid break on Chr2 and a dicentric chromosome involving Chr3 and Chr4 (*white asterisks*). *Right*, quantification of abnormal metaphases by M-FISH in control and TDP2 patient lymphoblastoid cells. Statistical significance was determined by GLM ANOVA (****P* < 0.005, NS, not significant). **c** Distribution of chromosome aberrations per abnormal metaphase. The total number of events (chromosome and chromatid exchanges/breaks) were counted in each abnormal metaphase (5 or more events per cell were pooled as > 5). Statistical significance was determined by ANOVA (**P* < 0.05, ****P* < 0.005)

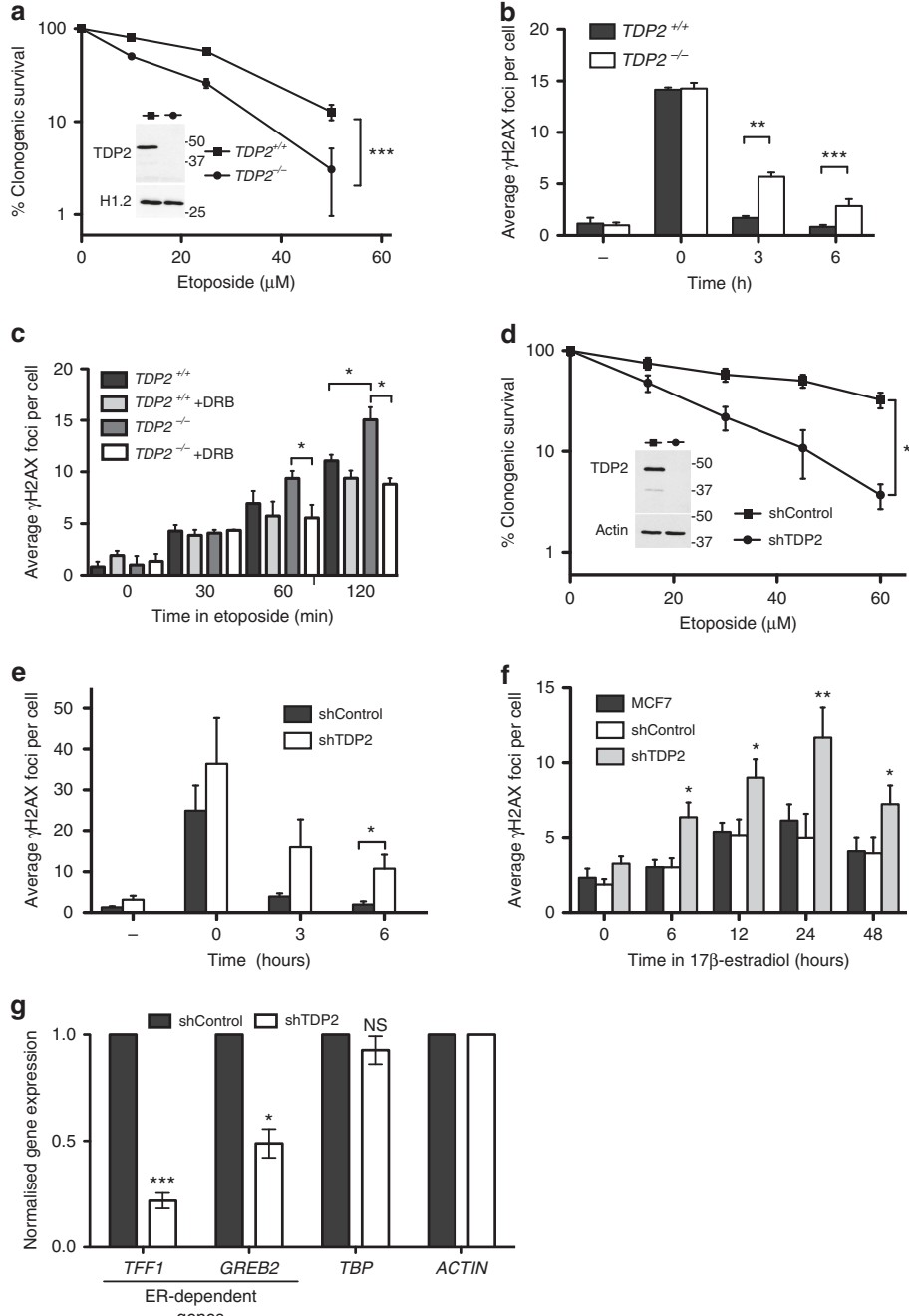

**Fig. 2** TDP2 protects gene transcription from TOP2-induced DSBs. **a** Clonogenic survival of wild-type and *TDP2*⁻/⁻ RPE-1 cells following treatment (3 h) with the indicated concentrations of etoposide. Data are the mean (±s.e.m.) of three independent experiments. Statistical significance was determined by two-way ANOVA (*P < 0.05, **P < 0.01, ***P < 0.005). Molecular weight markers are in KDa. **b** The number of γH2AX foci in wild-type and *TDP2*⁻/⁻ RPE-1 cells before and 30 min after treatment with 20 μM etoposide, and after the indicated repair periods in drug-free medium. Data are the mean ( ± s.e.m.) of at least three independent experiments. Statistical significance was determined by *T*-tests (*P < 0.05, **P < 0.01, ***P < 0.005). **c** The number of γH2AX foci in serum-starved wild-type and *TDP2*⁻/⁻ RPE-1 cells was measured following treatment with 5 μM etoposide. Where indicated, cells were pre-incubated with the transcription inhibitor DRB (100 μM) for 3 h prior to etoposide treatment. Other details are as in **b**. **d** Clonogenic survival of mock-depleted (shControl) or TDP2-depleted (shTDP2) MCF7 cells following treatment (3 h) with the indicated concentrations of etoposide. Other details are as in **a**. **e** The number of γH2AX foci in mock-depleted (shControl) or TDP2-depleted (shTDP2) MCF7 cells before and 30 min after treatment with 20 μM etoposide, and after the indicated repair periods in drug-free medium. Other details as in **b**. **f** The number of γH2AX foci in untransfected MCF7 cells or in mock-depleted (shControl) or TDP2-depleted (shTDP2) MCF7 cells following incubation for the indicated times with 100 nM 17β-estradiol. Data are the mean (±s.e.m.) of five independent experiments. Statistical significance was determined by *T*-tests (*P < 0.05, **P < 0.01) and are comparisons with shControl cells. **g** Basal mRNA levels of two ER-regulated genes (*TFF1* & *GREB2*) and one ER-independent gene (*TBP*) in mock-depleted (shControl) or TDP2-depleted (shTDP2) MCF7 cells. mRNA levels were quantified by qRT-PCR and normalized relative to *ACTB* levels under the same experimental conditions. The normalized value from TDP2 patient cells was then expressed relative to the normalized value from the mock-depleted control cells. Data are the mean (±s.e.m.) of three independent experiments. Statistical significance was determined by *T*-tests (*P < 0.05, ***P < 0.005, NS, not significant)

It is likely that cleavage complexes become abortive at some frequency even in the absence of TOP2 poisons. Indeed, TOP2-induced DSBs have been implicated in oncogenic translocation events not only in response to exposure to chemotherapeutic TOP2 poisons, but also in the formation of de novo translocations[5, 10, 15]. Consistent with this, cells possess DNA repair enzymes with activities targeted specifically at removing trapped topoisomerase activity from DNA 5′ termini, such as tyrosyl-DNA phosphodiesterase 2 (TDP2)[16–18]. TDP2 cleaves the phosphotyrosyl bond between the topoisomerase and the 5′ phosphate of the DNA generating ligatable ends that can be processed by the non-homologous end-joining (NHEJ) pathway[18, 19]. Here, we have addressed directly the impact of TDP2-dependent repair of TOP2-induced DSBs on genome stability and in chromosome translocations in human cells. We show that TDP2 rejoins DSBs induced during transcription-dependent TOP2 activity in breast cancer cells and at the translocation 'hotspot', *MLL*. Moreover, we find that TDP2 suppresses chromosome rearrangements induced by TOP2 and reduces TOP2-induced chromosome translocations that arise during gene transcription. Collectively, these data highlight the threat posed by TOP2 in genome instability during gene transcription and demonstrate the importance of TDP2-dependent NHEJ for suppressing this instability.

## Results

**Human TDP2 suppresses TOP2-induced genome instability.** We previously reported that TDP2-dependent NHEJ is required for genome stability in murine cells[18]. To determine whether TDP2 is required to maintain genomic stability in human cells we employed lymphoblastoid cells from a patient in which TDP2 is mutated[11]. These cells lack detectable TDP2 activity, are deficient in the repair of TOP2 breaks, and are hypersensitive to TOP2 poisons[11]. TDP2 patient cells exhibited a significantly greater increase in micronuclei and nucleoplasmic bridges than did normal control lymphoblastoid cell lines (LCLs) following treatment with low doses of etoposide, suggesting that genome instability is increased in the absence of TDP2 at sites of TOP2-induced DSBs (Fig. 1a). To measure the impact of TDP2 on chromosome stability directly, we conducted M-FISH of fixed chromosome preparations (Fig. 1b). Although similar numbers of abnormal metaphases were observed in control and TDP2 patient cells in the absence of etoposide treatment, treatment with an etoposide concentration that only slightly increased abnormal metaphases in control cells increased abnormal metaphases in TDP2 patient cells ~ fivefold; from 0.14 per metaphase to 0.68 per metaphase. Moreover, TDP2 patient cells harbored on average ~ 10-fold more chromosome aberrations per cell than did control cells following etoposide treatment (Supplementary Fig. 1; 3.96 and 0.4 aberrations per cell, respectively). These were comprised of both chromosome and chromatid-type events but the latter were far more prevalent; increasing ~ 30-fold in etoposide-treated TDP2 patient cells from 0.04 to 3.53 aberrations per cell, compared with little or no increase in normal cells (Supplementary Fig. 1). Notably, patient cell populations exhibited a particularly high frequency of metaphases with multiple aberrations, including complex events involving multiple chromosomes (Fig. 1c). Collectively, these results demonstrate that TDP2 is required to maintain chromosome stability in human cells following the induction of DSBs by abortive TOP2 activity.

**TDP2 protects gene transcription from TOP2-induced DSBs.** Although TDP2 can promote repair of TOP2-induced DSBs arising during S phase, we have proposed that the error-free

nature of this pathway may have particular utility in non-cycling cells that lack alternative error-free pathways such as homologous recombination repair[18]. Consequently, TDP2 may have particular utility in removing TOP2-induced DSBs arising during gene transcription, which can arise independently of cell cycle status. To address this, we employed diploid human RPE-1 hTERT cells in which we disrupted *TDP2* gene using CRISPR-Cas9. *TDP2*[−/−] RPE-1 cells exhibited hypersensitivity to etoposide and a reduced rate of repair of etoposide-induced DSBs, as measured using γH2AX immunofoci as a marker of DSBs[20] (Fig. 2a, b). Importantly, in quiescent cells, the elevated accumulation of etoposide-induced DSBs in *TDP2*[−/−] cells was prevented by pre-incubation with the RNA polymerase II transcription inhibitor DRB, suggesting that TDP2 is important for the repair of TOP2-induced DSBs arising during transcription (Fig. 2c).

TOP2 activity has been implicated previously in DSB induction during hormone-regulated gene transcription[9, 10]. Consistent with this, we showed previously that TDP2 is required to maintain normal levels of TOP2-dependent gene transcription stimulated by androgen[11]. To further address the role of TDP2 in the repair of DSBs induced during transcription we depleted TDP2 in the ER-positive breast cancer cell line MCF7 using anti-TDP2 shRNA. TDP2-depleted MCF7 cells exhibited reduced DSB repair following treatment with etoposide and correspondingly an increase in hypersensitivity to this drug (Fig. 2d, e). Interestingly, the addition of estradiol alone resulted in the accumulation of γH2AX foci in MCF7 cells, and did so to a greater extent in MCF7 cells depleted of TDP2 (Fig. 2f). More importantly, both the basal level and estradiol-stimulated level of transcription of *TFF1* and *GREB1*, two genes known to be regulated by estrogen, was reduced in TDP2-depleted MCF7 cells when compared to wild-type MCF7 cells (Fig. 2g and Supplementary Fig. 2). Collectively, these results implicate TDP2 in the removal of TOP2-induced DSBs arising during gene transcription and in the protection of transcription from inhibition by TOP2-induced DSBs.

**TDP2 protects *MLL* from TOP2-induced breakage.** Next, we investigated the impact of TDP2 on DNA breaks occurring at the *MLL* chromosome locus, which are also associated with TOP2 activity. Translocations involving the breakpoint cluster region (BCR) of *MLL* gene are amongst the most common chromosome rearrangements observed in leukaemia[7]. Rearrangements of the *MLL* gene can involve one of a large number of partners and are implicated in both therapy-related acute leukemia and infant acute leukaemia[21]. Numerous potential mechanisms for *MLL* translocation have been proposed, including nucleases involved in VDJ recombination, Alu-mediated recombination, apoptotic nucleases, and TOP2-induced breaks[7, 15]. To examine whether TDP2 is required to protect *MLL* from TOP2-induced breakage, we employed FISH probes flanking this locus so that we could detect etoposide-induced breaks via their impact on the spatial proximity of these probes (Fig. 3a).

The addition of etoposide increased to ~ 10% the fraction of wild-type RPE-1 cells harboring a visible break at one of the two *MLL* loci (Fig. 3a). More importantly, this fraction increased to ~ 30% in *TDP2*[−/−] RPE-1 cells, implicating TDP2 in the repair of TOP2-induced DSBs at *MLL*. Similar results were observed in lymphoblastoid cells, in which etoposide-induced more *MLL* breaks in TDP2 patient cells than in control cells (Fig. 3b). The breaks detected in these experiments were not apoptotic breaks[22], because we did not detect any evidence for significant apoptosis under the conditions employed (Supplementary Fig. 3). Importantly, the presence of elevated etoposide-induced breaks at *MLL*

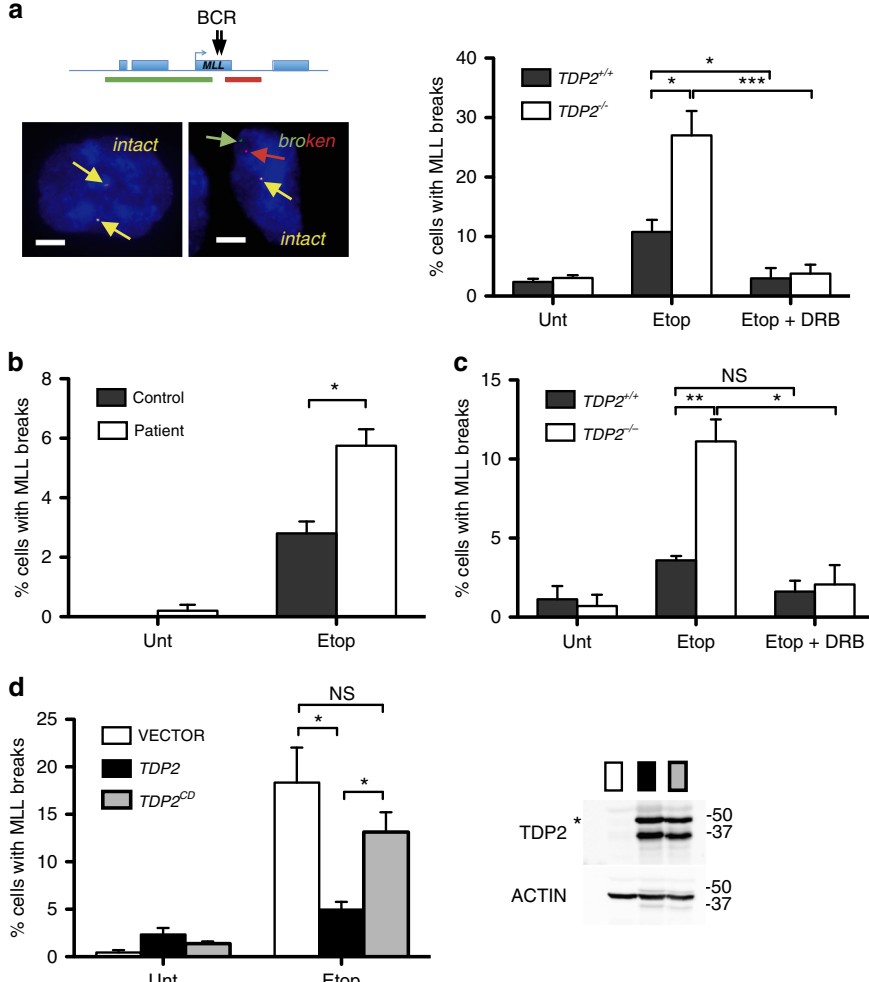

**Fig. 3** TDP2 is required for the repair of TOP2-induced DSBs at *MLL*. **a** Detection of broken *MLL* loci by FISH in wild-type and *TDP2*$^{-/-}$ RPE-1 cells following mock-treatment or treatment with 100 μM etoposide for 6 h. Where indicated, cells were pre-incubated with DRB (100 μM) for 3 h prior to etoposide treatment. *Left*, representative images of intact (*yellow*) and broken (*red* and *green*) *MLL* loci (*Scale bar* 5 μm). The position of the FISH probes (*red* and *green* lines in top cartoon) flanking the *MLL* breakpoint cluster are shown. *Right*, data are the mean (± s.e.m.) of three independent experiments. Statistical significance was measured by *T*-tests (*$P < 0.05$, **$P < 0.005$, ***$P < 0.001$). **b** Detection of broken *MLL* loci in normal and TDP2 patient lymphoblastoid cells, measured as above. Data are the mean (± s.e.m.) of two independent experiments. **c** Detection of broken *MLL* loci by FISH in serum-starved wild-type and *TDP2*$^{-/-}$ RPE-1 cells. Data as in **a**. **d** Detection of broken *MLL* loci by FISH in *TDP2*$^{-/-}$ RPE-1 cells infected with either empty lentiviral vector or vector expressing either wild-type TDP2 (*TDP2*) or catalytic mutant TDP2$^{D262A}$ (*TDP2*$^{CD}$). Data as in **a**. See inset for levels of TDP2 protein (full-length TDP2 is indicated by an asterisk). Molecular weight markers in KDa

in the absence of TDP2 was also observed in quiescent cells, consistent with the source of these breaks being TOP2 activity during transcription rather than DNA replication (Fig. 3c). Indeed, consistent with this, pre-incubation with the transcription inhibitor DRB greatly reduced or ablated the *MLL* breaks detected by FISH in both wild-type and *TDP2*$^{-/-}$ RPE-1 cells (Fig. 3a, c). The impact of DRB did not reflect an influence on cell cycle distribution[23], because we did not observe any impact of DRB in this respect (Supplementary Fig. 3). Finally, to confirm that the role of TDP2 in preventing *MLL* breakage required its catalytic activity we compared recombinant wild-type TDP2 and catalytic mutant TDP2[16, 17] for ability to complement this defect in *TDP2*$^{-/-}$ RPE-1 cells. Indeed, whereas wild-type TDP2 significantly reduced TOP2-induced *MLL* breakage, mutant TDP2 did not (Fig. 3d).

Collectively, these data indicate a role for TDP2 activity in the repair of TOP2-induced DSBs generated during gene transcription at the breakpoint cluster region of *MLL*; a chromosome locus that is a hotspot for oncogenic translocations.

**TDP2 suppresses TOP2-induced chromosome translocations**. To examine directly the impact of TDP2 on the frequency of TOP2-induced chromosome translocations we again employed FISH. Wild-type and *TDP2*$^{-/-}$ RPE-1 cells were maintained in G0/G1 during and following treatment with etoposide, to avoid the impact of TOP2-induced DSBs arising during DNA replication, and then released into cell cycle to enable the detection of translocations involving chromosome 8 or 11 by whole-chromosome FISH (Supplementary Fig. 4) Translocations were observed in both cell lines following etoposide treatment but were ~ 4 times more frequent in *TDP2*$^{-/-}$ RPE-1 cells, indicating that TDP2 suppresses the formation of chromosome translocations induced by TOP2 (Fig. 4a). Importantly, the translocations associated with loss of TDP2 were prevented by pre-incubation with DRB, implicating gene transcription by RNA polymerase II as the source of the TOP2-induced DSBs that resulted in translocation (Fig. 4a). Notably, expression of wild-type recombinant TDP2 but not the catalytic mutant TDP2 reduced the frequency of translocations in *TDP2*$^{-/-}$ cells (Fig. 4b). These results indicate

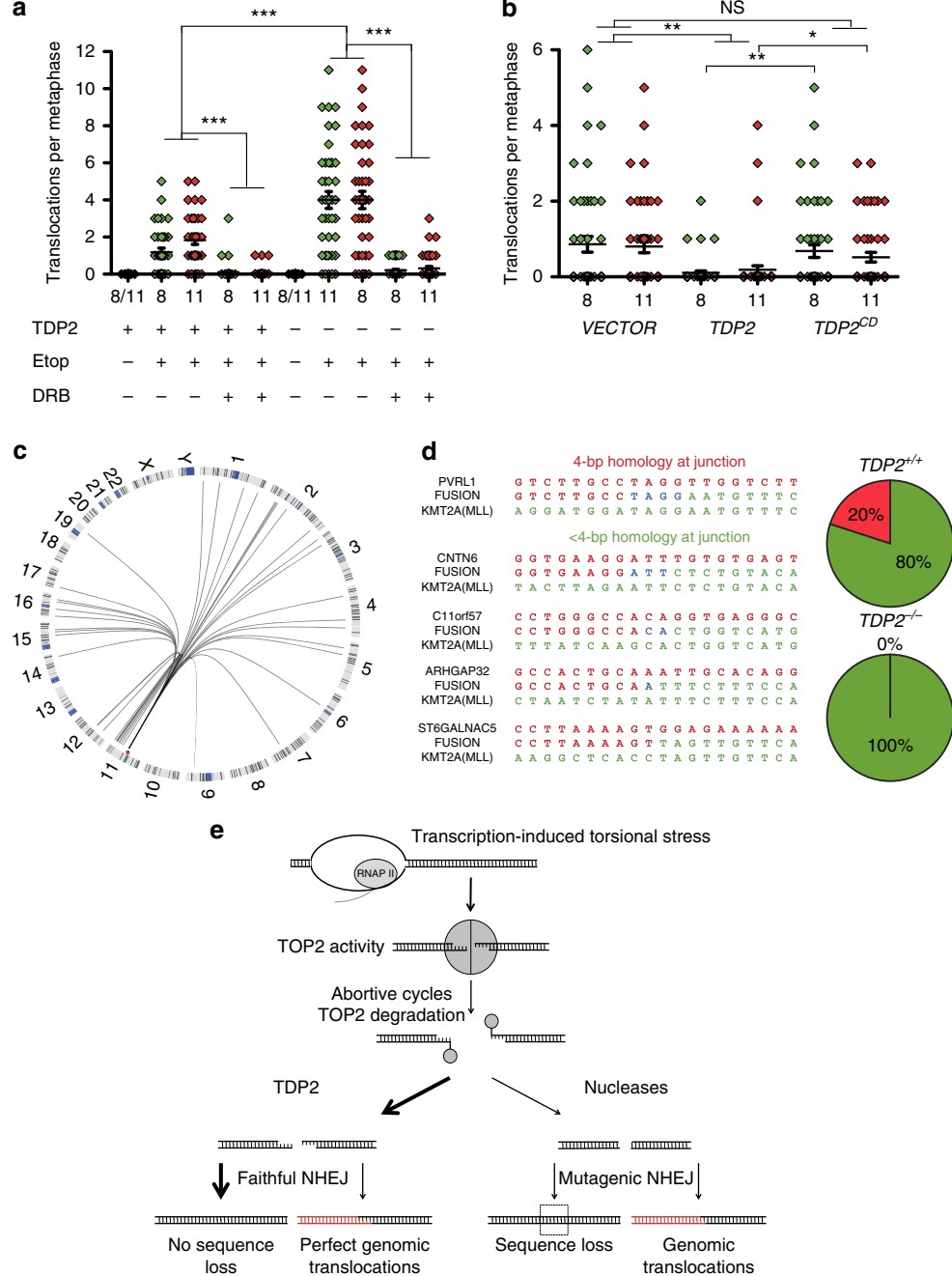

**Fig. 4** TDP2 suppresses TOP2-induced chromosome translocations. **a** Translocation frequencies (translocations/metaphase) in chromosomes 8 and 11 were quantified in serum-starved (G1/G0) cells in metaphase spreads prepared 48 hr after mock-treatment or etoposide treatment (1 h, 100 μM). In the absence of etoposide, translocations for both chromosomes are plotted together (8/11). Where indicated, cells were pre-treated with DRB (100 μM) for 3 h prior to, during, and 4 h after etoposide treatment to inhibit transcription. Data are from two independent experiments and statistical significance was measured by $T$-tests (***$P < 0.001$). **b** Translocation frequencies in etoposide-treated $TDP2^{-/-}$ RPE-1 infected with empty lentiviral vector or with lentiviral vector encoding wild-type TDP2 or catalytic mutant TDP2[D262A] ($TDP2^{CD}$). Data are from two independent experiments and statistical significance was determined by $T$-tests (*$P < 0.05$, **$P < 0.005$, NS, not significant). **c** Circos plot showing the genomic distribution of *MLL* translocation partners in etoposide-treated wild-type RPE-1 cells. The location of *MLL* in chromosome 11 is indicated (*red*). Resolution; cytogenetic bands. **d** *Left*, representative junction sequences of etoposide-induced *MLL* translocations with 4-bp (*top*) or <4-bp regions of junction homology (*bottom three*). *Right*, distribution of translocation junctions with 4-bp or < 4-bp regions of homology in WT and $TDP2^{-/-}$ RPE-1 cells. **e** Model depicting the influence of TDP2-dependent and independent NHEJ on translocations at TOP2-induced DSBs during transcription. *Left*, TDP2 primarily promotes rejoining of the correct (adjacent) termini (*left*, thick arrow) but occasionally may promote joining of incorrect (distal) termini with the same TOP2 cleavage sequence (*left*, thin arrow). We propose that the latter events are rare but result in a translocation with 4-bp of junction homology. *Right*, in the absence of TDP2, processing of 5′-phosphotyrosine termini requires nuclease activity and consequently error-prone NHEJ resulting in microdeletions (if the correct adjacent termini are rejoined) or translocations (if incorrect distal termini are joined) with ≠ 4-bp of junction homology

that TDP2 activity suppresses TOP2-induced translocations during gene transcription.

To analyze the nature and sequence of the translocations induced by TOP2 in the presence and absence of TDP2 we employed inverse PCR to amplify the breakpoint cluster region (BCR) of *MLL* (Supplementary Fig. 5). Etoposide treatment increased the appearance of PCR products that were different in size from that expected for wild-type product, suggesting that we had successfully isolated translocation breakpoints (Supplementary Fig. 5). Of a total of forty-three different *MLL* translocation breakpoints that we isolated, 84% involved partner loci that were located in actively transcribed regions, consistent with transcription stimulating the *MLL* translocations (Supplementary Table 2). Interestingly, 26% of the *MLL* translocation partners were present in the same or adjacent cytogenetic chromosome bands as *MLL* (Fig. 4c). This is in agreement with the notion that physical proximity greatly influences the choice of translocation partner[3]. We failed to detect a significant variation between wild-type and *TDP2*[−/−] cells in the distribution of *MLL* partners, suggesting that TDP2 does not influence the site of TOP2-mediated cleavage (Fig. 4c and Supplementary Table 2). Intriguingly, however, Sanger sequencing of the translocation junctions suggested that whereas some translocation events in wild-type cells involved 4 bp of identical sequence at the breakpoint in both *MLL* and the translocation partner, these events were absent in *TDP2*[−/−] cells (Fig. 4d). Thus, while TDP2 overall greatly suppresses chromosome translocations at sites of TOP2 cleavage, it might be responsible for a small subset of translocations that are characterized by 4-bp of perfect homology at the translocation junction.

## Discussion

In this study we have addressed the impact of TDP2-dependent DSB repair on chromosome rearrangements mediated by the abortive activity of TOP2. M-FISH revealed that lymphoblastoid cells from a TDP2-defective patient incur elevated chromosome aberrations following the induction of TOP2-induced DNA breaks by etoposide. These results are in agreement with previous cytogenetic analyses of lymphoid tissue in etoposide-treated *Tdp2*[−/−] mice[18]. Chromatid-type breaks were particularly prevalent in etoposide-treated TDP2 patient cells, consistent with the high level of TOP2 activity associated with chromosome duplication during S/G2-phase[24].

Another major source of genome instability and tumorigenesis are DSBs that arise during gene transcription, which can result in chromosome translocations. However, the origins of these DSBs and the molecular processes by which they result in translocations are unclear. Here, we have shown that TDP2 suppresses transcription-associated DSBs induced by TOP2 and prevents the accumulation of DSBs arising during estrogen stimulation. It is currently unclear whether the estrogen-induced breaks in TDP2-defective cells reflected estrogen-stimulated transcription or estrogen-stimulated entry into S phase. However, the former possibility is consistent with the requirement for TDP2 for normal levels of androgen-driven transcription in prostate cancer cells[11]. Indeed, in the current study, TDP2-depleted cells exhibited reduced expression of estrogen-regulated genes in breast cancer cells, suggesting that TDP2 does indeed protect estrogen-regulated gene transcription from inhibition by TOP2-induced DSBs.

Strikingly, TDP2 also prevented the accumulation of etoposide-induced DSBs within the breakpoint cluster region of the mixed lineage leukemia (*MLL*) locus. These observations have important implications for carcinogenesis because TOP2-induced DSBs are strongly implicated in oncogenic translocations[15], including those

arising at *MLL*[10, 15]. Moreover, based on whole-chromosome FISH of chromosome 8 and 11, our results suggest that TDP2 suppresses the frequency of etoposide-induced chromosome translocations genome-wide. Importantly, the translocation events that were suppressed by TDP2 were almost entirely transcription dependent. These data highlight both the threat posed by abortive TOP2 activity to genome stability during transcription and the importance of TDP2-dependent DSB repair in suppressing TOP2-induced genome instability. It should be noted, however, that we did not detect an impact of TDP2 on genome instability in the absence of etoposide. This is consistent with the similar lack of genome instability in *Tdp2*[−/−] mice, in the absence of etoposide treatment. The apparent absence of genome instability in *TDP2*[−/−] cells and mice at endogenous levels of TOP2-induced DNA breakage most likely reflects the availability of alternative mechanisms for removing TOP2 from DSB termini, such as Mre11-dependent NHEJ and/or HR[25].

Sequence analysis of the translocation junctions at *MLL* in our experiments revealed that ~ 80% of the translocation partners were transcriptionally active regions. Moreover, there was strong bias towards translocation partners located on the same chromosome as *MLL*, consistent with the proximity of TOP2-induced DSBs influencing the likelihood of rejoining incorrect chromosome termini. Since TDP2 activity generates DSB termini that are ligated by the canonical pathway for cNHEJ[18], our discovery that TDP2 supresses the overall frequency of etoposide-induced chromosome aberrations and translocations implicates a protective role for NHEJ events involving TDP2 in genome stability. This is in contrast to DSBs induced by recombinant nucleases, which in human cells implicate NHEJ in the formation of chromosome translocations[26]. We propose that this difference reflects the nature of the termini at the DSBs. In contrast to most DSBs, those induced by TOP2 and processed by TDP2 result in 4-bp cohesive 5′-overhangs that enable the rapid ligation of adjacent termini without further processing, thereby reducing the chance of joining together incorrect termini[19]. This idea may also explain our observation that a small subset of translocation junctions involving 4-bp of perfect homology appeared to be absent in *TDP2*[−/−] cells. We suggest that while adjacent 4-bp cohesive DSB termini generated by TDP2 are normally correctly re-ligated, on rare occasions these termini re-ligate incorrectly with a distal TOP2-induced DSB harboring an identical 4-bp sequence overhang. This type of junction has been reported for a therapy-related *MLL* translocation in leukemia[27, 28], and suggested to arise by TOP2 domain swapping[21, 29, 30]. While further work is required to conform this observation, our data suggest that this class of translocation may result from TOP2-induced DSBs processed by TDP2 (Fig. 4e). It is important to note, however, that most of the TOP2-induced translocations arising in wild-type and *TDP2*[−/−] RPE-1 cells either lacked homology at the junction or harbored smaller regions of microhomology, consistent with previous observations[31] and with their generation by TDP2-independent inaccurate NHEJ.

In summary, transcriptionally active loci are regions of the genome commonly associated with translocations[32], and often these loci are associated with high levels of TOP2 activity[9, 10, 12]. Our data demonstrate the threat posed by TOP2-induced DSBs to gene transcription and genome stability, and highlight the role of TDP2-dependent NHEJ in protecting transcription and genome stability from these breaks.

## Methods

**Cell culture**. Normal and TDP2 patient[11] LCLs were maintained in RPMI supplemented with 10% FCS and with penicillin and streptomycin. RPE-1 cells (originally purchased from ATCC) were propagated in DMEM/F12 medium supplemented with 10% FCS. *TDP2*[−/−] RPE-1 cells expressing wild-type human

TDP2 or catalytic mutant TDP2$^{D262A}$ (TDP2$^{CD}$) were generated using the lentiviral vector *pSIN-DUAL-GFP*. Cells were sorted for GFP positivity before use, and all experiments were performed in cultures with at least 70% of GFP-positive cells. For serum starvation cells were grown until confluency and then cultured in 1% FCS for 5–8 days. All cell lines were grown at 37 °C, 5% CO$_2$. For ER-dependent gene induction cells were washed with serum-free medium three times for 30 min and incubated in medium containing 5% charcoal-stripped FBS for at least 48 h before experiment. Etoposide, 17β-estradiol (estradiol) and 5,6-dichlorobenzimidazole 1-β-D-ribofuranoside (DRB) were purchased from Sigma. All cell lines were tested for mycoplasma contamination.

**Western blotting**. Anti-TDP2 antibody was employed at 1:5000 in western blotting and has been described previously[33]. Anti-Actin (Sigma A4700) was employed at 1 : 2000 and anti-Histone 1.2 (Abcam ab17677) at 1 : 20,000. Full blots are included in Supplementary Fig. 6.

**Generation of TDP2-depleted MCF7 cells and TDP2$^{-/-}$ RPE-1 cells**. To stably deplete TDP2, MCF7 cells were co-transfected with pcD2E vector and either pSuper-TTRAP/TDP2 or pSuper as described previously[16] and selected in 0.5 mg ml$^{-1}$ G418 to isolate single clones. To delete TDP2, we employed CRISPR-Cas9[34]. A 17-bp (minus the PAM) Tru-guide[35] RNA sequence targeting TDP2 exon 1 (5′-GCGGCGACTTCTGTGTG-3′) was selected using the E-CRISP tool E-CRISP (http://www.e-crisp.org/E-CRISP/) and cloned into the guide RNA vector #41824 (AddGene). Briefly, the Tru-guide sequence minus the PAM were annealed and extended into a 98-mer double-stranded fragment using Phusion polymerase (NEB) which was then subcloned into the guide RNA vector using Gibson Assembly (NEB)[36]. The TDP2 guide construct was co-transfected with hCas9 expressed from plasmid #41815 (AddGene) using a Neon electroporation system (Invitrogen). Transfected cells were enriched by selection in 0.5 mg/ml G418 (Thermofisher) for 5 days prior to isolation of single clones and screening for loss of TDP2 expression by western blotting.

**Clonogenic survival assays**. Cells were plated in 10 mm plates and 3 h later treated with indicated concentrations of etoposide for 1 h. Cells were rinsed twice with PBS and incubated with fresh drug-free media for 10 days and then fixed in 70% ethanol/1% methylene blue. The surviving fraction at each dose was calculated by dividing the average number of colonies (>50 cells) in treated dishes by the average number in untreated dishes.

**Micronuclei and nucleoplasmic bridges**. To detect micronuclei (MN) and nucleoplasmic bridges (NB), cytochalasin B (Sigma) was added to cell cultures at 4 mg ml$^{-1}$. After 24 h, cells were fixed and stained with DAPI.

**Metaphase spreads**. For metaphase spreads, cells were incubated with demecolcine (Sigma) at 0.2 μg/ml for 4 h and then harvested. Cells were collected using standard cytogenetic techniques and fixed in 3 : 1 methanol:acetic acid. Fixed cells were dropped onto acetic acid-humidified slides.

**Immunofluorescence and FISH analysis**. For immunofluorescence (IF), cells were grown on coverslips for 2 days (for cycling cultures) or 8 days (for serum-starved and confluency-arrested cell cultures) and then treated as indicated. Cells were fixed (10 min in PBS-4% paraformaldehyde), permeabilized (2 min in PBS-0.2% Triton X-100), blocked (30 min in PBS-5% BSA), and incubated with the indicated primary antibodies for 1–3 h in PBS-1% BSA. Cells were then washed (3 × 5 min in PBS-0.1% Tween 20), incubated for 30 min with the corresponding AlexaFluor-conjugated secondary antibody (1 : 1000 dilution in 1% BSA-PBS) and washed again as described above. Finally, cells were counterstained with DAPI (Sigma) and mounted in Vectashield (Vector Labs). γH2AX foci were manually counted (double-blind) in 20 to 40 cells per data point per experiment. For the analysis of G0/G1 cells, only CENP-F negative cells were scored. Primary anti-γH2AX (Millipore, 05-636) were employed at 1 : 1000 and anti-CENP-F (Abcam, ab5) at 1 : 500. FISH was performed according to manufaturer's protocol (CytoCell). For M-FISH, a modified version of Metasystems protocol was used. Chromosome and chromatid aberrations were analyzed and categorized as exchanges or breaks. Exchange type aberrations were further categorized as 'simple' (involving maximum of two breaks in two chromosomes) or 'complex' (any exchange involving >3 breaks)[37].

**Inverse PCR**. For the detection of single chromosomal translocations an adaptation of an inverse PCR protocol[38] was performed. After treatment, genomic DNA was extracted with the DNeasy Blood and Tissue Kit (Qiagen). 10 μg of DNA were incubated for 1 h with Shrimp Alkaline Phosphatase (NEB). After inactivation and digestion with either Bgl2 or Sac1 (NEB) DNA was purified with QIAquick PCR Purification Kit (Qiagen). To ensure ligation in *cis* of DNA molecules DNA was diluted at 18 μg/ml and ligated for 20 h at 4 °C. After purification and quantification 40ng of DNA were used per PCR reaction. Two consecutive PCR reactions (LongAmp polymerase, NEB) were performed to ensure specific amplification. First-round primers: MLLBgl2_1: (5′-

GAGGAAATCAGCACCAACTGGGGGAA-3′) and MLLBgl2_2: (5′-GATCCTGTGGACTCCATCT GCTGGAA-3′) or MLLSac1_1 (5′-ATATGGGTGCAAAGCACTGTAT-3′) and MLLSac1_2 (5′-ACCAGTCCTTCAACTTCTGGG-3′). One microliter of a 1 : 10 dilution of the first-round PCR mix was amplified with second-round primers: MLLBgl2_3: (5′-CATTAGCAGGTGGGTTTAGCGCTGGG-3′) and MLLBgl2_4: (5′-TTCTCCTGCTTATTGACCGGAGGTGG-3′) or MLLSac1_3 (5′-CAGTA-GACCCCTGGCACTTG-3′) and MLLSac1_4 (5′-AGCAATCTCACAGGGT TCCT-3′). Novel-size bands were purified from agarose gel and sequenced using MLLBgl2_4 or MLLSac1_5 (5′-AGTGCAGTGGCGTGATAATG-3′) primers. Translocations were represented as by a Circos plot[39] using the OmicCircos package of R.

**Data availability**. All data are available from the authors on request.

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

## Acknowledgements

We thank F. Cortés-Ledesma for providing TDP2 lentiviral vectors, A. Rodríguez-Gil for his help with Circos plots, and M.J. Castro-Pérez for FACS technical support. F.G.-H. was funded by a CR-UK Programme grant to KWC (C6563/A16771) and by the Spanish Ministry of Economy and Competitivity (RYC-2014-16665 and BFU2016-76446-P). M.I.M.-M. and G.Z.V. were funded by a CR-UK Programme grant to KWC (C6563/A16771). A.H.-R. was recipient of a Leonardo da Vinci Vocational Education and Training program (MERCURIO 2013-1-ES1-LEO02-66587).

## Author contributions

All experiments were performed by F.G.-H. unless indicated. G.Z.-V. and I.N. generated *TDP2*$^{-/-}$ RPE-1 cells and TDP2-depleted MCF7 cells, respectively. G.Z.-V., I.N., and M.I.M.-M. conducted clonogenic survival assays, γH2AX assays, and transcription assays. R.M.A. performed and scored M-FISH on LCLs. A.H.-R. performed MLL break-apart FISH on LCLs. K.W.C. and F.G.-H. conceived the study, designed the experiments, interpreted the results, and wrote the manuscript.
