## [Peer Review File · Nature Communications]

Reviewers' comments:

Reviewer #1 (Remarks to the Author):

This short paper by Gomez-Herreros and Caldecott demonstrate the critical role of TDP2 to prevent genomic damage and translocations. The impact of TDP2 on etoposide is expected, but the role of transcription is novel. Most interesting and novel are the role of TDP2 in the estrogen-induced gene regulatory pathways and in oncogenic translocations, which implies the involvement of TOP2 in those alterations.

The paper is clearly written and concise. The data appear solid.

Our few editorial/technical comments are:

1. Statistical analyses and number of independent repeats need to be added to Figs. 3B (right panel: effect of DRB on Etop-induced MLL breaks) and Fig. 2G (comparison of shControl vs. shTDP2 with estradiol).
 2. The recent publication by Hoa, N.N. et al. 2016 Mol Cell should be discussed (line 254).
 3. Regarding the data presented in Fig. 4C, the authors may also discussed an prior study that sequenced TOP2 sites and recombination induced by amsacrine (Han H-Y., Povirk, L. 1993 JMB).
 4. Discussion: lines 262-3: the statement "TDP2 is part of the canonical pathway for Ku-dependent non-homologous end-joining (NHEJ)" could be misunderstood. "Classical" end-joining acts independently of TDP2. Please clarify.
- Line 276: TOP2 domain swapping (subunit exchanges): the most relevant references are Ikeda, H et al. 1982 PNAS and Pommier, Y et al. 1985 Cancer Res.

Reviewer #2 (Remarks to the Author):

This paper addresses in detail the consequences of TDP2 loss on the response to TOP2 poisons. The focus is on the types of genomic instability induced by TDP2 loss in human or mouse cells exposed to TOP2 poisons. Data is presented in support of the idea that TDP2 acts on both replication-related and transcription-related TOP2-induced lesions, where it suppresses both unrepaired breaks (e.g. chromatid breaks) and genome rearrangements. Although some of the early data in the m/s appears partly confirmatory of previously published data on TDP2, the data on MLL translocations and, especially, the breakpoint sequencing is new and convincing. The authors provide compelling data linking TDP2 activity to the generation of 4nt microhomologous rearrangement breakpoints in TOP2-poisoned cells. The breakpoint analysis also strongly supports a contribution of transcription-associated breaks in MLL BCR rearrangements. In further support of a role for transcription-induced breaks as initiating lesions of genomic instability, DRB is shown to suppress indices of genomic instability in TOP2-poisoned TDP2^{-/-} cells. However, it would be helpful to formally determine whether DRB affects cell cycle progression in these experiments. Similarly, estradiol is known to promote proliferation of MCF7 cells. In this regard, the induction of breaks in estradiol-treated MCF7 cells could reflect a proliferative effect of this hormone—i.e., replication-related breaks rather than transcription-induced breaks. The m/s would be strengthened by analysis of the cell cycle status of the different treatment groups in these experiments.

It would be ideal to rescue/complement the breakage/instability phenotype of the TDP2^{-/-} patient cells with wtTDP2, since the comparison made in these experiments is not between isogenic cell lines. This concern is mitigated by that fact that the key data, presented later in the m/s, derives from work in isogenic TDP2^{-/-} vs. wtRPE cells.

Specific points

Some figure font sizes make the text illegible – need to improve figure presentation.

Fig 2—panel E caption is missing

Fig S2—contributes little to the m/s. The image seems anecdotal.

Fig S3—FACS plots look uncompensated – region boundaries appear arbitrary

Fig S5—define red arrows

Line 91 – define LCLs

Line 107: “these results demonstrate that TDP2 is required to maintain chromosome stability in human cells following the induction of DSBs by abortive TOP2 activity.” This is a confirmation of previously published data and should be identified as such.

Does DRB treatment affect the cell cycle in the experiments shown?

Does estradiol treatment affect the cell cycle status of MCF7 cells in the experiments shown?

It would be helpful to examine other DNA damaging agents in some of the breakage/genomic instability assays to establish the specificity of the effects observed for etoposide.

Line 212: “TDP2 is responsible for a specific subset of chromosome translocations characterized by 4 bp of conserved homology at the translocation junctions.” The term “microhomology” is a widely accepted term to describe this breakpoint type.

There are a few typos throughout the m/s.

Reviewer #3 (Remarks to the Author):

Gomez-Herreros and colleagues report that TDP is required to suppress chromosomal translocations during transcription. They show that, in the absence of TDP2, caused either through knockdown of TDP2 or in TDP2 mutant patient cells, levels of DSB caused by impaired TOP2 activity are increased. They extend this finding to show that TOP2 is required for the repair of TOP2-generated DSB after hormone driven gene expression in the breast cancer derived cell line MCF7. Failure to repair TOP2-associated breaks results in elevated levels of translocation. Looking specifically at MLL translocation the authors report that a proportion of these translocations are probably linked to TOP2 since they are also suppressed by TDP2. Moreover these translocations are linked to damage that occurs during transcription rather than replication as it occurs in non-cycling cells. Finally the authors sequence the break-join region of the translocations to show that a significant proportion of the TOP2 associated translocations in TDP2-defective cells differ from those produced in wild type cells, having lost a signature 4bp region of homology present in MLL and translocation partner that characterizes translocations from TDP2 proficient cells.

This is an interesting and well-conceived study, which makes a strong case for the role of TOP2-induced breaks and their TDP2-dependent repair in chromosome translocation. The experiments are well performed and analysis of the data supports the overall conclusions. The findings add further evidence for the important interplay between DNA repair and transcription that has implications for cancer and other diseases.

Specific points: -

In the results section there are various references to 5-fold, 30-fold and 90-fold differences in specific aspects of the data from different cell lines, but it was not clear exactly which data/values the authors were comparing to infer these differences. It would be helpful to make this clearer.

The majority of the study compares TDP2 defective cells (often a TDP null) with wild type. It is

possible that absence of TDP2 affects other members of a repair complex which might not assemble when TDP2 is absent. To be sure that the observations made are due to the activity of TDP2 and not caused by indirectly simply through absence of TDP2 protein, the authors should complement the defect (in at least one assay) with TDP2 mutated for its catalytic activity. Ideally they might do this in their translocation assay where they can sequence the re-joined TOP2 dependent breaks. This would show that the altered break-join structure is linked directly to TDP2 tyrosyl-DNA phosphodiesterase activity.

Reviewer #1

1. Statistical analyses and number of independent repeats need to be added to Figs. 3B (right panel: effect of DRB on Etop-induced MLL breaks) and Fig. 2G (comparison of shControl vs. shTDP2 with estradiol). *Response; statistical analysis has now been added in all cases, and the number of independent repeats for each experiment is indicated in the legend. In Fig.2G, experimental variation between experiments in the level of gene induction by estradiol prevented the difference between shControl and shTDP2 from being statistically significant, despite this difference being apparent in all of the individual replicates (i.e. the same trend in each experiment). In order to improve this, we therefore measured transcription levels of these genes in shControl and shTDP2 cells at steady-state, in the absence of estrogen. This approach worked, with clear differences between gene transcription in the absence and presence of TDP2 that were statistically significant. This new data is included in Fig.2G, with the old Fig.2G now in Fig.S2.*
2. The recent publication by Hoa, N.N. et al. 2016 Mol Cell should be discussed (line 254). *Response; This has now been added (Page 8).*
3. Regarding the data presented in Fig. 4C, the authors may also discussed an prior study that sequenced TOP2 sites and recombination induced by amsacrine (Han H-Y..Povirk, L. 1993 JMB). *Response; this reference has now been added as part of an extended discussion on page 9.*
4. Discussion: lines 262-3: the statement “TDP2 is part of the canonical pathway for Ku-dependent non-homologous end-joining (NHEJ)” could be misunderstood. “Classical” end-joining acts independently of TDP2. Please clarify. *Response; Clarification is added to the discussion, page 8 and 9.*
5. Line 276: TOP2 domain swapping (subunit exchanges): the most relevant references are Ikeda, H et al. 1982 PNAS and Pommier, Y et al. 1985 Cancer Res. *Response; These references have now been added (page 8).*

Reviewer #2

1. It would be ideal to rescue/complement the breakage/instability phenotype of the TDP2^{-/-} patient cells with wtTDP2, since the comparison made in these experiments is not between isogenic cell lines. This concern is mitigated by that fact that the key data, presented later in the m/s, derives from work in isogenic TDP2^{-/-} vs. wtRPE cells. *Response; Complementation of patient cells LCLs is extremely difficult, so instead we have complemented the TDP2^{-/-} RPE-1 cells with wild-type and catalytic-dead TDP2. This new data is presented in Fig. 3D. and Fig. 4B.*

Specific points

2. Some figure font sizes make the text illegible – need to improve figure presentation. *Response; Thank you for addressing this point. We will adapt them to Nat. Comms font size requirements.*
3. Fig 2—panel E caption is missing *Response; this is now included.*
4. Fig S2—contributes little to the m/s. The image seems anecdotal. *Response; We agree and we have now removed it.*
5. Fig S3—FACS plots look uncompensated – region boundaries appear arbitrary. *Response; Indeed, the FITC signal was not compensated. However, this does not affect the results. As the PI signal is minimal, QR and QL can be considered together,*

and compensation does not change these values. Boundaries were adjusted to positive control (+CPT).

6. Fig S5—define red arrows. *Response; definition has been added to the legend of Supplementary Figure 5.*

7. Line 91 – define LCLs. *Response; definition of LCLs (lymphoblastoid cells) has been added in results and methods.*

8. Line 107: “these results demonstrate that TDP2 is required to maintain chromosome stability in human cells following the induction of DSBs by abortive TOP2 activity.” This is a confirmation of previously published data and should be identified as such. *Response; We have now added a statement at the beginning of the indicated paragraph (page 3) summarising this previous work (which was in mouse cells). “We previously reported that TDP2-dependent non-homologous end-joining is required for genome stability in murine cells (Gomez Herreros, 2013)”.*

9. Does DRB treatment affect the cell cycle in the experiments shown? *Response; We have examined the impact of DRB on cell cycle profile in our experiments and we do not see any significant differences (new Figure S3A). However, to further rule out an impact of cell cycle on our results we have also used serum-starved (arrested) cells in Fig. 3C. DRB abolished the MLL breakage in these non-cycling cells, supporting that transcription inhibition rather than cell cycle kinetics underlie the suppression of MLL breakage by DRB. We have included this data and discuss it in the new version of the manuscript (page 5). In the case of our measurements of genomic translocations, all DRB treatments were in arrested cultures.*

10. Does estradiol treatment affect the cell cycle status of MCF7 cells in the experiments shown? *Response; because estradiol promotes both transcription and progression into S-phase it is not possible to separate these events. However, we have now also measured the level of basal transcription of the same genes in the absence of estrogen (new Fig.2g). These experiments confirm the importance of TDP2 for normal levels of TOP2-dependent gene transcription.*

11. It would be helpful to examine other DNA damaging agents in some of the breakage/genomic instability assays to establish the specificity of the effects observed for etoposide. *Response; We have previously shown that the sensitivity and repair defects observed in TDP2-lacking cells are restricted to TOP2-poisons, and are not evident following other sources of DNA damage such as ionising radiation or MMS (Gómez-Herreros, et al. 2013, Gómez-Herreros, et al. 2014).*

12. Line 212: “TDP2 is responsible for a specific subset of chromosome translocations characterized by 4 bp of conserved homology at the translocation junctions.” The term “microhomology” is a widely accepted term to describe this breakpoint type. *Response; we agree with the reviewer but we would prefer to avoid the word ‘microhomology’ in this context to prevent misunderstanding with the ‘alternative NHEJ’ pathway that is also known as microhomology-mediated. TDP2 Is not part of this pathway, but rather functions as part of the ‘classical’ NHEJ pathway (Gomes-Herreros et al 2013).*

Reviewer #3

1. In the results section there are various references to 5-fold, 30-fold and 90-fold differences in specific aspects of the data from different cell lines, but it was not clear exactly which data/values the authors were comparing to infer these differences. It

would be helpful to make this clearer. *Response; thank you – we have now corrected some of these figures and clarified which comparison each quantification refers to.*

2. The majority of the study compares TDP2 defective cells (often a TDP null) with wild type. It is possible that absence of TDP2 affects other members of a repair complex which might not assemble when TDP2 is absent. To be sure that the observations made are due to the activity of TDP2 and not caused by indirectly simply through absence of TDP2 protein, the authors should complement the defect (in at least one assay) with TDP2 mutated for its catalytic activity. Ideally they might do this in their translocation assay where they can sequence the re-joined TOP2 dependent breaks. This would show that the altered break-join structure is linked directly to TDP2 tyrosyl-DNA phosphodiesterase activity. *Response; We have used a lentiviral expression system in order to complement MLL breakage and genomic translocations after etoposide treatment with wild type and catalytic-dead TDP2. With this result we now demonstrate that the repair of transcription-dependent breaks at the MLL translocation and that TDP2 transcription-induced genomic translocations depend on TDP2 activity. Please find new results in Fig.3D and Fig.4B. Although we like the idea of similarly repeating the approach for sequencing translocation junctions, this was not feasible in the time period available to us. We hope the referee and editor understand our position.*

REVIEWERS' COMMENTS:

Reviewer #1 (Remarks to the Author):

The authors have addressed my comments. This is an important study.

Reviewer #2 (Remarks to the Author):

The concerns raised previously have been addressed and I recommend acceptance of the m/s.

Reviewer #3 (Remarks to the Author):

The authors have added the essential experiment suggested. And seem to have answered the majority of reviewers' criticisms satisfactorily.